# Characterization and Design of Circular Binders

Hans C. Hendrikse [1],*, Hamza El Khallabi [2], Thomas Hartog [1], Aikaterini Varveri [3] and Anthon Tolboom [1]

1 Latexfalt B.V., 2396 AP Koudekerk aan den Rijn, The Netherlands
2 Institute for Life Sciences & Chemistry, Hogeschool Utrecht, 3584 CS Utrecht, The Netherlands
3 Pavement Engineering, Engineering Structures Department, Delft University of Technology, 2628 CN Delft, The Netherlands; a.varveri@tudelft.nl
* Correspondence: hans.hendrikse@latexfalt.com

**Abstract:** The concept of a circular economy, where waste materials are transformed into valuable resources, is gaining increasing attention. However, many waste streams are difficult to recycle due to their mixed composition and broad molecular distribution. This paper explores the potential of repurposing mixed materials, specifically focusing on creating a circular alternative to bitumen, a fossil-based binder used in road construction. The molecular weight and composition of bitumen are analyzed using gas chromatography (GC) and infrared spectroscopy (IR). This study proposes using waste plastics and bio-based oils to develop a paving binder with similar molecular distribution. Various plastic types, such as low-density polyethylene (LDPE), high-density polyethylene (HDPE), isotactic polypropylene (PP), polystyrene (PS) and polyethylene terephthalate (PET), are examined for their compatibility with different oils. It is observed that the compatibility of both the molecular weight and composition between the plastic and oil is crucial for the successful dissolution and homogeneity of the binder. Additionally, the crystallinity of the plastic plays a role in the flexibility and durability of the resulting binder. It is demonstrated that by carefully selecting waste materials and understanding their molecular characteristics, it is possible to create circular alternatives to fossil-based materials like bitumen. This approach has the potential to reduce waste, lower dependence on fossil resources, and contribute to sustainable and circular construction materials.

**Keywords:** bitumen alternative; circular binder; gas chromatography; waste plastic; tall oil pitch; molecular distribution



## 1. Introduction

In an ideal circular economy, every waste stream is utilized to fabricate a new material with economic value [1,2]. Significant efforts are therefore being made to recycle materials, so they can be re-used for their original purpose [3–6]. However, this horizontal recycling is often not fully possible. According to PlasticsEurope, currently, only 30% of all waste plastic is recycled, while the rest ends up in landfills or is incinerated for energy [7]. This is due to most waste streams being either mixed in composition or having a broad molecular distribution, making them unsuitable for use as virgin materials. It would greatly benefit the circular economy if there was an outlet for these mixed materials which still would have economic value, without resorting to incineration or landfilling.

In this context, repurposing materials will be a necessity to prevent waste and create a true circular economy. Yet, repurposing of leftover components is not a new development, and has already been done in the old 'throw-away' society to make a profit from these components. A prime example of this is bitumen, which is the leftover residue after refining crude oil [8]. It is primarily used as a binder in road construction, where 80 million tons of it is processed into asphalt roads annually [9].

Being the residue of oil distillation, its chemical composition is complex and can vary widely depending on the origin of the bitumen. Since its main function is to bind aggregate and maintain acceptable stiffness over a wide temperature range, its characteristics are

often described using rheological measurements [10,11]. What is known from a chemical perspective, however, is that bitumen is comprised of a mix of hydrocarbon, which traditionally have been classified by the SARA analysis [12]. Over time, infrared spectroscopy (IR) has complemented this technique and has given more insight into the specific chemical components present [13]. Moreover, as the product of a distillation process bitumen has quite a high molecular weight, where the lower cutoff is determined by the distillation process which can be shown with gas chromatography (GC) [14].

A circular alternative to bitumen, which has similar properties but is made from non-fossil compounds, could be an excellent outlet for waste streams too mixed for other purposes: it stores the $CO_2$ into a material that has a lifetime of at least 12 years and is often recycled. At the same time, it could make the road construction industry less dependent on its 80 million tons of fossil material [15–19]. Moreover, this alternative bitumen should consist of a mixture of hydrocarbons with a broad molecular distribution, making the mixed nature of the waste streams an interesting prospect.

In this study, a method is developed to characterize a bitumen on its molecular weight and functional groups. This approach can be extended to waste streams so direct comparison between both material types can be made. Next, the method is utilized to incorporate plastics commonly found in household waste in conjunction with other circular materials to produce a circular alternative to bitumen. The method is then used to analyze problematic characteristics that can occur while developing a binder in this fashion. From the lessons learned, a circular binder with similar properties to bitumen is produced, but made from the waste streams of society.

## 2. Materials and Methods

Materials and Sample Preparation. Bitumen presented in this paper are samples of bitumen used for production at Latexfalt B.V. over the past 10 years. The reference oil used in this research is a mix of various aromatic extender oils obtained from Eni Versalis, Repsol and Valochem. The tall oil pitch (TOP) was obtained from Kraton. The circular C5-resin (Eastotac H-142R) was obtained from Synthomer. The soybean oil was obtained from Cargill. The rapeseed methyl ester was obtained from Vandeputte Oils & Oleochemicals. Waste stream household plastics were obtained from Umincorp and HVC. To examine pure plastics, plastics from common household objects were used: HDPE was obtained from sandwich bags, LDPE was obtained from foils, and PP and PS were obtained from plastic cups. Plastics were mixed into the various oils at 180 °C using a Disperlux mixer. Typically, 5–20 wt% of plastic was dissolved, which took usually between 1–3 h.

Analysis Methods. Gas chromatography measurements were done according to an internal method with a Trace 1300 from Thermo Scientific using a FID detector and an MXT-1HT Sim Dist column of 5 m length, 0.52 cm thick and a packing of 0.1 μm. Measurements were started at an oven temperature of 90 °C, which ramped to 430 °C with a rate of 10 °C/min. The resulting measurements were normalized in order to compare them to one another.

Infrared spectroscopy measurements were performed according to an internal method with a Shimadzu IR Spirit. For every measurement, a background signal was measured to minimize fluctuations. Samples were applied to the sample holder at room temperature; 20 scans per measurement were performed and no further processing was done on the retrieved signal.

Viscosity measurements were done according to an internal method with a Thermo Scientific Rheostress 1 parallel plate viscometer. Measurements were started at 180 °C, after which the temperature was lowered to 100 °C in steps of 10 °C/min. During each step, 3 min of equilibration time was used.

Cohesion measurements were done with a Petrotest DDA 3 according to NEN-EN 13589 at 5 °C.

Penetration measurements were done with a Petrotest Penetrometer PNR10 according to NEN-EN 1426.

Softening point Measurements were with a PAC ISL RB365G according to NEN-EN 1427.

## 3. Results and Discussion

### 3.1. Molecular Weight and Composition of Bitumen

Bitumen is the leftover product of oil distillation. As such, one can expect that it is not a specific molecule nor does it have a certain molecular weight. Instead, it has a more or less continuous molecular distribution which has a specific minimum weight: the point where the distillation process was stopped. This can be well identified with gas chromatography (GC) which runs from 90 °C to 450 °C (Figure 1A). Instead of an individual peak indicating a single molecule being present, it shows more of a landscape of molecules with various boiling points.

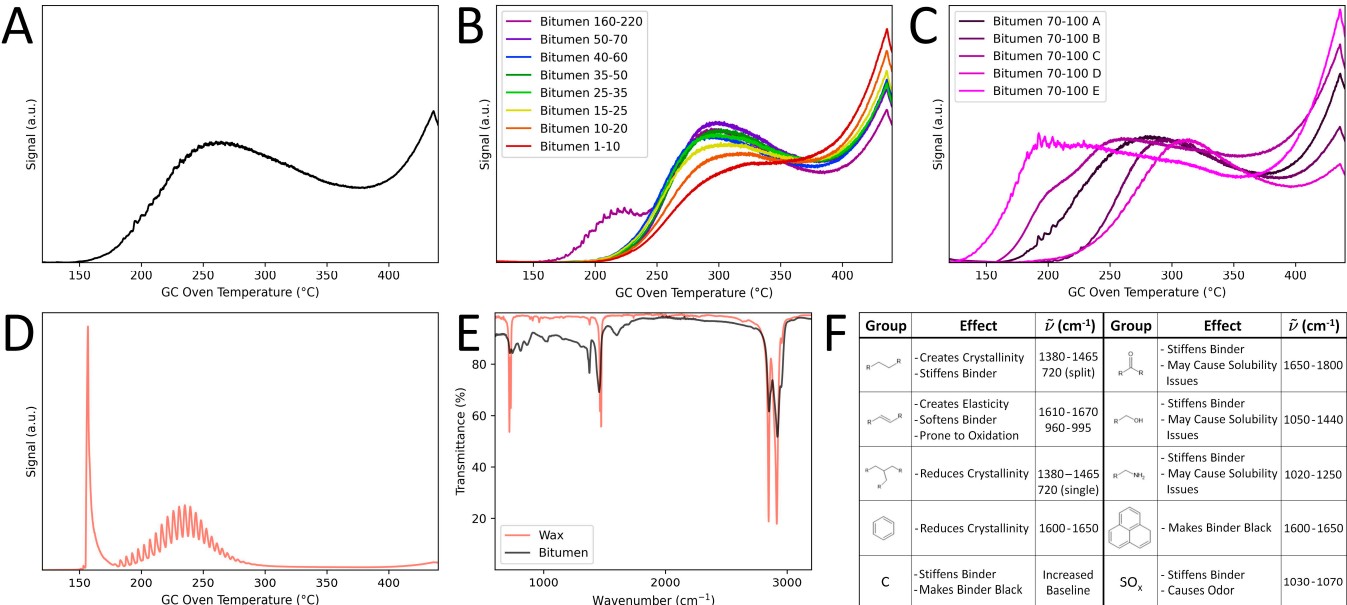

**Figure 1.** Classifying a bitumen based on its molecular weight and composition. (**A**) Typical GC-chromatograph of a bitumen, showing its wide molecular distribution with various boiling points. (**B**) The average molecular weight of a bitumen determines its pen grade. (**C**) Bitumen of the same pen grade, but different supplier, can vary widely in molecular weight distribution but has comparable average molecular weight. (**D**) GC-chromatograph of a wax, which has a significantly lower boiling point compared to the bitumen shown in (**A**). (**E**) IR of the wax of (**D**) and bitumen of (**A**). The twin peaks at 720 cm$^{-1}$ and 1470 cm$^{-1}$ show that the wax is predominantly made of non-branched aliphatic hydrocarbons, whereas bitumen comprised a mix of hydrocarbons indicated by the various visible peaks. (**F**) A collection of common groups found in bitumen, along with their effect on the stiffness of the binder and the wavenumber at which they can be detected with IR.

There are multiple classes of bitumen which are identified by their 'pen-grade': a measure of the stiffness of a bitumen which is measured by penetrating a sample with a needle. Interestingly, one can relate this stiffness to the molecular distribution similar to how the stiffness of a polymer is related to its molecular weight: the higher the average molecular weight, the stiffer the polymer. For bitumen, this means that its lower molecular weight components, which come from the GC earlier due to a lower boiling point, contribute to a softer pen grade, whereas the higher molecular weight components cause a stiffer pen grade. In this view, bitumen can be seen as a hydrocarbon polymer, it just has a very wide distribution of molecular weight and composition. The pen grade can therefore be seen as an indication of the average molecular weight of this 'polymer'.

The relation between the average molecular weight and the pen grade can be well seen when a single crude bitumen of the same supplier, but with different pen grades, is

analyzed with GC (Figure 1B). Here, it is plain to see that the harder grades (indicated by a lower number, for example, 1–10 in red) contain more high molecular weight components, whereas the softer grades contain in comparison more low molecular weight components (for example, 160–220 in purple).

Though this pen grade has proven its effectiveness by providing proper roads for decades now, chemically speaking a pen grade can be achieved in a plethora of ways as it does not say anything about the molecular weight distribution. Figure 1C shows GC chromatograms of various bitumen with the same (70–100) pen grade, which varies significantly in their molecular weight distribution. Bitumen E starts its signal at a lower temperature compared to bitumen D for example, indicating lower molecular weight components with a lower boiling point being present. And even though the GC cannot detect components with a boiling point above 430 °C, it is clearly visible Bitumen E does contain a higher amount of detectable higher molecular weight components compared to bitumen D, making the average molecular weight of both bitumen similar. Thus, a pen grade can be made with a wide variety of molecular distributions, as long as their molecular weight averages out to a similar weight.

Simply looking at the GC to determine molecular weight based on boiling point is not enough to determine the stiffness of a binder. Even though bitumen is predominantly composed of hydrocarbons, there are other atoms like oxygen, nitrogen and sulfur present and the organization of the hydrocarbons (linear aliphatic, branched aliphatic, aromatic and saturated or unsaturated) has a significant effect on the stiffness, which can be identified with infrared spectroscopy (IR). Figure 1D for example shows a wax, whose average boiling point is significantly lower than that of bitumen. However, its typical application in a binder is to stiffen it. This is because its melting point is quite high compared to its boiling point as its aliphatic chains can easily crystallize (Figure 1E), thus making it a solid, stiff material in a binder. For bitumen, this difference in boiling point and melting point averages out because a large mix of various hydrocarbons is present. When a circular equivalent binder is made from specific materials this is not the case, however; thus, the general function of a particular functional group in a binder needs to be known. In general, if a functional group causes crystallinity or polar interactions, it will stiffen the binder, whereas groups that prevent these interactions will soften the binder [20–22]. A summary of common hydrocarbon groups found in bitumen and their expected behavior is given in Figure 1F.

*3.2. Circular Binder Design Principles*

With this, the principle of making a circular equivalent of bitumen is set: this binder should contain materials whose combined molecular distribution is wide and whose functional groups are compatible with one another. Potentially this can be done with waste materials from society; for example, bio-based oils with a relatively high boiling point for the low molecular weight part and waste plastics which cannot be recycled horizontally for the high molecular weight part. For the plastics, common household plastics consisting of low-density polyethylene (LDPE), high-density polyethylene (HDPE), isotactic polypropylene (PP), polystyrene (PS) and polyethylene terephthalate (PET) were examined. For the low molecular weight part, oils with a relatively high boiling point were used, particularly soybean oil (a byproduct of soybean protein production), methyl esters of rapeseed oil, tall oil pitch (a leftover product from the paper industry) and a fossil-based aromatic extender oil for reference. In Figure 2A an overview of the miscibility of these materials is given. Rather than going into the details of each mixture, we will focus on general trends that were observed while creating these mixtures.

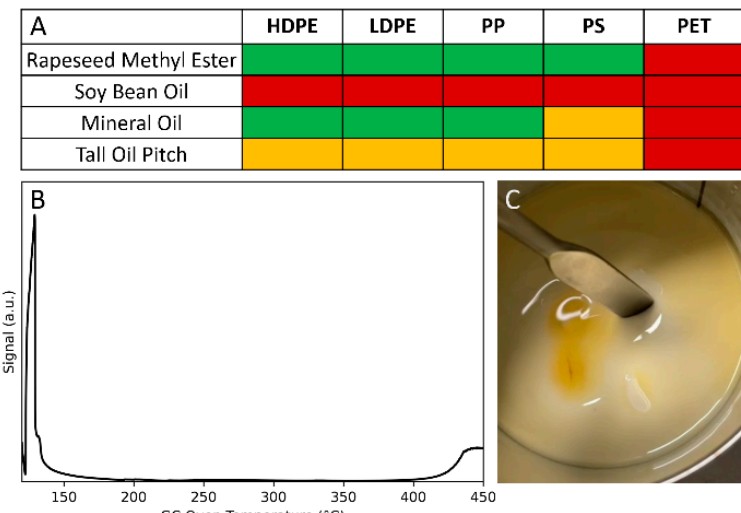

**Figure 2.** Effect of the choice of oil on the binder. (**A**) Overview of solubility of plastics in various oils. Green indicates full solubility, orange indicates solubility but with inhomogeneous stiffness and red indicates insolubility. (**B**) GC chromatogram showing that there is a gap in molecular distribution between the used oil (rapeseed methyl ester, RME) and plastic (polystyrene, PS) of this binder. (**C**) As soon as the binder of RME and PS is agitated, the oil behaves as a liquid whereas the plastic behaves as a solid, causing phase separation due to the large discrepancy in molecular weight between the two components.

First off, it is important that the molecular weight between the oil and the plastic is not too different. Using oils with only low molecular weight components creates a discrepancy in the molecular distribution. This can be clearly seen when rapeseed methyl ester is used, as this oil only has low molecular weight components which come from the GC at roughly 140 °C (Figure 2B). At first glance, its low molecular weight might actually seem beneficial when trying to dissolve the plastics (Figure 2A), as the smaller molecules can more easily get in between the plastic to start the dissolution process. However, a binder made of PS and RME has a discrepancy in its molecular weight distribution as can be seen in the GC (Figure 2B). Though this is not a problem while the binder is at rest, as soon as the binder is agitated with a spatula, the RME will react as a liquid and the PS as a solid, causing phase separation between the two (Figure 2C). This makes the binder unsuited for withstanding the load of traffic, and thus as a material for road construction.

A second precaution that should be taken, is to ensure that the plastic used in the binder is not too crystalline. For example, if a binder is made with exclusively HDPE (Figure 3A), the linear polyethylene chains can stack easily into crystalline domains as shown in Figure 3C. These chains have their covalent bonds all in the same direction, making them tough when force is applied perpendicular to this direction as indicated in Figure 3C with a green arrow. However, in the parallel direction of these chains, the material is only held together with van der Waals interactions. When force is applied in that direction (indicated with a red arrow), the chains can relatively easily be separated. This forms brittle domains, which on a macro scale can be seen in the binder: simply putting a finger on the binder causes it to break (Figure 3E). As bitumen is the flexible part of a road and road failure usually occurs when the bitumen becomes too stiff to fulfill this role, introducing this brittleness in a fresh binder will be deleterious to the performance of the road from the get-go.

However, it is possible to circumvent this brittleness. If a binder is made with LDPE (Figure 3B), the polyethylene contains side branches that interfere with the crystalline domain formation as shown in Figure 3D. Here, the side branches create covalent bonds in all directions, so the binder can resist forces from all sides as indicated by the green arrows. On a macro scale, a binder is created which is flexible as shown in Figure 3F. Moreover, it is

even possible to create a flexible binder that contains HDPE, as long as LDPE is also added to break the crystal domains of the HDPE. Thus, the fact that these binders potentially will be made with mixed plastics is actually a positive prospect, as the mixing of molecules helps prevent the formation of brittle crystalline domains.

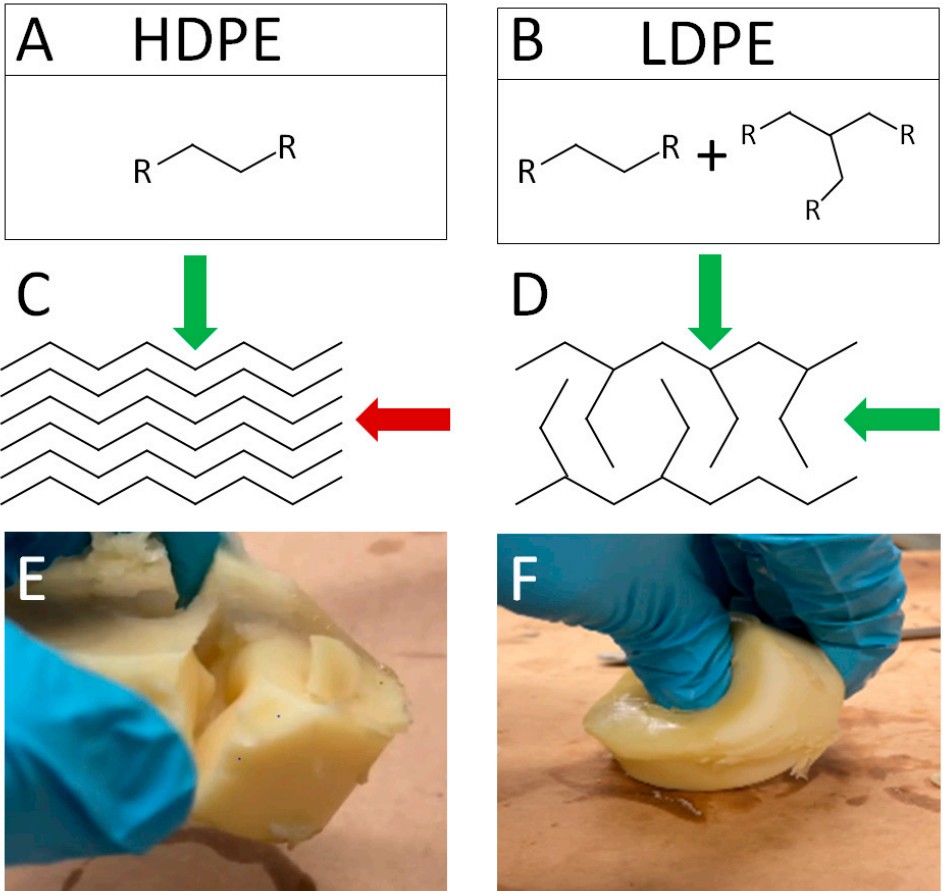

**Figure 3.** Effect of the choice of plastic on the binder. (**A**) HDPE comprised linear aliphatic hydro-carbons. (**B**) LDPE is also comprised of aliphatic hydrocarbons, but has side branches in its chains. (**C**) HDPE stacks its linear chains. Force applied parallel to these chains (indicated with a red arrow) easily breaks them up. (**D**) The side branches of LDPE prevent them stacking, so that they can resist forces from all sides (indicated by the green arrows). (**E**) Simply placing a finger on a binder with only HDPE as the high molecular weight component will cause it to crack. (**F**) Introducing LDPE to the binder will create a flexible binder, even if it contains some HDPE.

### 3.3. Circular Binder from LDPE and TOP

When taking into account the molecular distribution and need for amorphous poly-mers, a binder visually very similar to bitumen can be created (Figure 4E). This binder uses a tall oil pitch, which has a broad molecular distribution to prevent phase separation (Figure 4B) as the transition from liquid to solid occurs more gradually in a material with such a molecular distribution. For the high molecular weight components, the aforemen-tioned LDPE is used to introduce as little crystallinity as possible. These components are still a bit away from each other in terms of molecular weight and polarity, making them not fully compatible with one another, which results in the binder becoming stiffer on the top compared to the bottom as the plastics agglomerate at the top (Figure 2A). However, by adding a broad molecular C5-resin from post-consumer waste (Figure 4B) to help bridge the molecular weight gap and mineral oil to lower the polarity of the liquid phase (Figure 4A), it is possible to create a homogenous binder similar to bitumen.

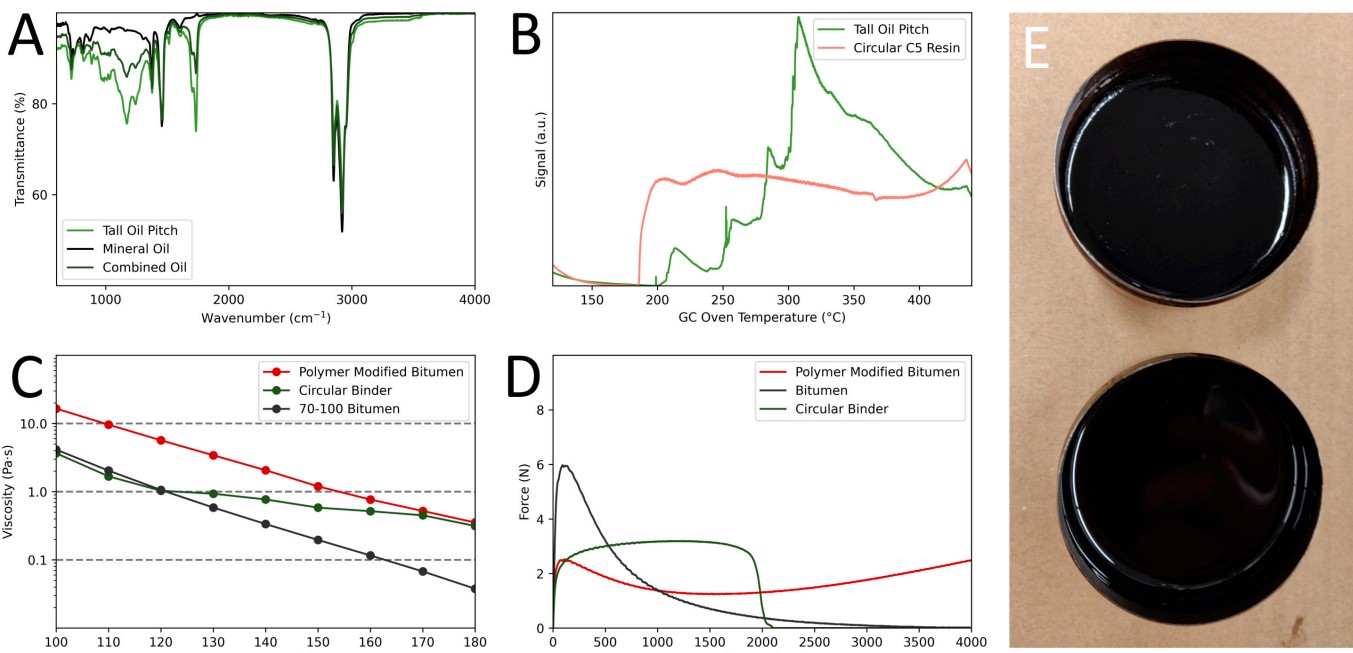

**Figure 4.** Circular binder based on tall oil pitch and LDPE. (**A**) IR spectrum of mineral oil (black) and tall oil pitch (light green). By combining both oils (dark green), the polar groups of the oil phase indicated by the peaks between 1050–1440 cm$^{-1}$ and 1650–1800 cm$^{-1}$ can be reduced to make the apolar LDPE better soluble. (**B**) GC of tall oil pitch and the circular C5 resin. The broad molecular weight distribution of both components indicated by their wide boiling point creates a binder that does not show phase separation upon agitation. (**C**) Viscosity of bitumen (black), polymer-modified bitumen (red) and the circular binder (dark green). At temperatures below 120 °C the circular binder has similar viscosity to normal bitumen, whereas its viscosity slowly becomes more like a polymer-modified bitumen as the temperature increases to 170 °C. (**D**) Cohesion at 5 °C of a 70–100 bitumen (black), the circular binder (dark green) and a polymer-modified bitumen (red). (**E**) Visual example of the circular binder (**top**) and bitumen (**bottom**).

Though this binder is similar to bitumen in looks, its properties are not identical to it. For example, it has a pen of 74, similar to a 70–100 bitumen. Yet, its softening point is 112 °C, significantly higher than that of 70–100 bitumen (43–51 °C) and even of a polymer-modified bitumen (PMB, softening point ranging up to 100 °C depending on polymer modification percentage). Moreover, its viscosity at temperatures above 120 °C is higher than a regular bitumen, due to the plastic's high molecular weight keeping a steady baseline viscosity around 1 Pa·s (roughly the viscosity of glycerol at room temperature, Figure 4C). However, at temperatures below 120 °C this baseline becomes similar to bitumen, as the continuous molecular distribution of the bitumen starts to have more and more components in the solid phase. Thus, at temperatures below 120 °C, the new binder actually has a similar viscosity to bitumen, giving it an interesting prospect to be used as a low-temperature binder. Currently, the asphalt industry is transitioning more and more to low-temperature asphalt production because of the lower production energy needed and reduced emissions [23], and this binder could be well integrated into this transition with its viscosity properties.

When looking at the cohesion performance of the material (Figure 4D), it is also apparent that the binder behaves differently from normal bitumen: Bitumen quickly loses its cohesive strength when pulling it apart, whereas the circular binder keeps its cohesion for a significantly greater elongation. In fact, it looks like it is somewhere in between the performance of bitumen and a polymer-modified bitumen, which intuitively makes sense as the high molecular components of this binder are in fact polymers. The PMB can sustain even longer elongation before losing its cohesion, as it has both the high molecular weight components of bitumen as well as a tailored polymer (styrene–butadiene–styrene co-block

polymer, SBS) that contribute to its cohesive strength. The fact that the circular binder can imitate these properties with cheaper waste plastics at similar processing temperatures makes it very appealing for the circular economy.

With these observations, it becomes apparent that when designing a binder via the method described here, it becomes possible to tune the properties of said binder. Though in this paper only one example of a bitumen alternative is given, many more suitable combinations of waste materials must exist. Finding the proper selection of materials will not only enable us to keep re-using these materials after they are no longer suited for their original purpose; it also opens up a plethora of possibilities for new road construction design as the composition of materials can be tuned to specific needs.

## 4. Conclusions

In this work, bitumen has been described based on its molecular weight and composition using GC and IR analysis, respectively. From this, a general principle of how bitumen should be built up has been established, which has been applied to make circular binders based on waste plastics and bio-based oils. The evaluation of these circular binders has highlighted the significance of the molecular weight distribution of the selected oil and the level of crystallinity in the selected plastics. These factors greatly influence the properties of the binders, emphasizing the importance of careful waste selection to achieve desired performance characteristics. By adhering to the established design principles, a circular alternative to bitumen was developed using a combination of low-density polyethylene (LDPE) and tall oil pitch (TOP). The properties of this binder were analyzed and exhibited similarities to bitumen, indicating its potential as a sustainable substitute in road construction applications.

This study serves as a foundation for the design of new binders using circular waste streams. Future research directions could explore additional analytical techniques such as gel permeation chromatography for the identification of larger molecular weight components beyond the detection range of GC. Additionally, a combined GC-IR approach could be employed to analyze the molecular composition across different molecular weight ranges. Furthermore, the design principles presented in this study are not limited to waste plastics and bio-based oils; they can be extended to incorporate other materials like rubber or pyrolysis oils. In doing so, these design principles could find a binder with an optimal combination between performance, cost and environmental impact, offering exciting prospects for the development of circular binders.

In summary, this research provides a framework for the design and development of materials derived from mixed waste streams. Embracing the mixed nature of waste streams enables the road construction industry to actively contribute to the circular economy while introducing innovative materials with diverse applications. The principles established in this study pave the way for future advancements in the field, holding great potential for the realization of a more sustainable and circular future.

**Author Contributions:** Conceptualization, H.C.H.; Data Curation, H.C.H. and H.E.K.; Formal Analysis, H.C.H. and H.E.K.; Funding Acquisition, A.T.; Investigation, H.E.K.; Methodology, H.C.H. and H.E.K.; Project Administration, H.E.K.; Resources, A.T.; Supervision, H.C.H.; Validation, T.H. and A.V.; Visualization, H.C.H.; Writing—Original Draft, H.C.H.; Writing—Review and Editing, T.H. and A.V. All authors have read and agreed to the published version of the manuscript.

**Funding:** This research received no external funding.

**Data Availability Statement:** Data is contained within the article.

**Acknowledgments:** The authors would like to thank Peter Rem of the TU Delft for his valuable insights and assistance in obtaining various recycled plastic. They would also like to thank the R&D department of Latexfalt B.V. for providing assistance in the experiments. Finally, they would like to thank Benjamin Broeze of the Hogeschool Utrecht for supervising the internship of H.E.K.

**Conflicts of Interest:** The authors declare no conflict of interest. The funders had no role in the design of the study; in the collection, analyses, or interpretation of data; in the writing of the manuscript; or in the decision to publish the results.

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
