# Peer review of "Characterization and Design of Circular Binders"

_sustainability, doi:10.3390/su151712853_

Round 1

Reviewer 1 Report

The authors aim to obtain a binder similar with bitumen using recycled materials as plastic and bio-based oils. It is presented a methodology to identify the composition of the bitumen based on the molecular weight of the components which was further applied experimentally to obtain a binder with similar properties.

However, some clarifications are needed:

Page 104, Section 3.1. Please explain the need for comparison between the bitumen reference sample and the wax sample. The two materials are mixed in the final product, i.e. the circular binder?

If the collection of common groups found in bitumen and given in Figure 1F are identified by the authors in correlation with Figure 1E, it should be better explained in the article text.

Some minor corrections:

Line 207: Correct the word „HPDE” with „HDPE”.

Line 384: [19] isn’t a bibliographic reference but a note/statement which may be included in the article text.

Best regards.

Reviewer 2 Report

This is a well-prepared paper; however, here are some suggestions for improving the manuscript:

1. The concept of a circular binder should be further defined, along with its limitations and practicality.

2. A more related literature review should be provided in Section 1 and within the result's discussion to support the findings or argument.

3. Material properties and characteristics should be provided.

4. Figure 4(E) should be replaced with a better image so that the reader can appreciate their similarities or differences based on observation.

5. It would be great if the authors could relate to the cost, energy, and carbon emission aspects.

6. Please revise references 8 and 19.
